# GraphSite: Ligand Binding Site Classification with Deep Graph Learning

**DOI:** 10.3390/biom12081053

**Published:** 2022-07-29

**Authors:** Wentao Shi, Manali Singha, Limeng Pu, Gopal Srivastava, Jagannathan Ramanujam, Michal Brylinski

**Affiliations:** 1Division of Electrical and Computer Engineering, Louisiana State University, Baton Rouge, LA 70803, USA; wshi6@lsu.edu (W.S.); eejaga@lsu.edu (J.R.); 2Department of Biological Sciences, Louisiana State University, Baton Rouge, LA 70803, USA; msing21@lsu.edu (M.S.); gsriva2@lsu.edu (G.S.); 3Center for Computation and Technology, Louisiana State University, Baton Rouge, LA 70803, USA; lpu1@lsu.edu

**Keywords:** structure-based drug discovery, ligand binding sites, deep learning, graph neural network

## Abstract

The binding of small organic molecules to protein targets is fundamental to a wide array of cellular functions. It is also routinely exploited to develop new therapeutic strategies against a variety of diseases. On that account, the ability to effectively detect and classify ligand binding sites in proteins is of paramount importance to modern structure-based drug discovery. These complex and non-trivial tasks require sophisticated algorithms from the field of artificial intelligence to achieve a high prediction accuracy. In this communication, we describe GraphSite, a deep learning-based method utilizing a graph representation of local protein structures and a state-of-the-art graph neural network to classify ligand binding sites. Using neural weighted message passing layers to effectively capture the structural, physicochemical, and evolutionary characteristics of binding pockets mitigates model overfitting and improves the classification accuracy. Indeed, comprehensive cross-validation benchmarks against a large dataset of binding pockets belonging to 14 diverse functional classes demonstrate that GraphSite yields the class-weighted F1-score of 81.7%, outperforming other approaches such as molecular docking and binding site matching. Further, it also generalizes well to unseen data with the F1-score of 70.7%, which is the expected performance in real-world applications. We also discuss new directions to improve and extend GraphSite in the future.

## 1. Introduction

Proteins carry out numerous biological functions in the cellular environment. Interactions between proteins and other molecules, such as peptides, neurotransmitters, nucleic acids, hormones, lipids, and metabolites, are, therefore, vital to understanding the biology of the cell. In particular, interactions between proteins and small molecules, or ligands, are associated with a wide range of the functions of a living cell [1]. Ligand binding sites are typically pockets and cavities on the surface of proteins formed by spatially close amino acid residues interacting with small molecules in a specific way [2]. The ability to precisely detect and annotate these sites in protein structures is of paramount importance in modern structure-based drug discovery. It can help reveal novel targets for pharmacotherapy and support the design of biopharmaceuticals not only against the most common health issues affecting a large population worldwide [3] but also rare diseases without any treatment options currently available [4]. Numerous approaches have been developed over past years to identify and analyze ligand binding sites in proteins, including LIGSITE [5], FTSite [6], *e*FindSite [7], Fpocket [8], and SiteComp [9], to mention a few examples. A comprehensive characterization of ligand binding accounts for multiple factors of this multifaceted phenomenon, such as the conformational dynamics [10], the druggability [11], interaction hotspots [12], and the amino acid composition [13]. Despite the encouraging progress in ligand binding site detection, there is a need for a better functional characterization of the identified sites with respect to the types and properties of binding molecules.

It has been demonstrated that similar ligands can bind to evolutionary unrelated proteins [14]. Therefore, accurate methods to classify binding sites depending on the ligand information are essential to study ligand binding at a system level with a broad range of applications in polypharmacology [15], side effects prediction [16], and drug repositioning [17]. Several algorithms to predict binding sites in protein targets, given the ligand information, have been developed to date. For instance, the ProBiS-ligands web server can help figure out the types of ligands binding to the input protein structures [18]. As many ligands perform specific cellular functions important for a variety of biological processes, such as cell signaling, active transport, cell metabolism, and the regulation of the cell cycle, several algorithms focus on specific types of ligands. VitaPred employs the evolutionary information to predict residues interacting with vitamin ligands [19], SITEPred identifies nucleotide-binding residues from protein sequences [20], and HemeBIND detects heme binding residues based on the sequence and structure information [21]. Similar techniques were designed to work with other specific organic molecules, such as flavin adenine dinucleotide [22], guanosine triphosphate [23], nicotinamide adenine dinucleotide [24], and inorganic ions, such as calcium [25] and zinc [26]. Most of these methods employ traditional machine learning classifiers to predict binding residues based on the sequence, structure, and evolutionary information. However, currently available state-of-the-art deep learning approaches hold significant promise to greatly improve the accuracy of the functional annotation of ligand binding sites.

Deep learning is currently the most advanced group of machine learning techniques employing various types of multilayer artificial neural networks to learn complex patterns from the input data. Deep learning makes headway in the computer vision field, where it has successfully been applied across numerous tasks, including object detection [27], face recognition [28], and body pose estimation [29]. A key to the success of deep learning methods is the convolutional neural network (CNN), which utilizes local trainable filters to effectively learn hierarchical latent features from the Euclidean data, such as 2D and 3D images [30]. Advances in computer vision have inspired the development of deep learning tools for biology and biomedicine as well. Most approaches to predict and annotate ligand binding sites in proteins with CNNs represent pockets as either 2D or 3D images. The former group of methods includes BionoiNet, which first projects pockets onto a 2D plane encoding various physicochemical, structural, and evolutionary properties, and then employs a 2D-CNN to perform classification tasks [31]. An example of a 3D-based approach is DeepDrug3D, which deploys a 3D-CNN to accurately classify binding sites for adenosine triphosphate (ATP) and heme ligands represented as voxel-based 3D images [17]. A related method, DeeplyTough, employs a similar pocket representation as DeepDrug3D and pocket matching with a CNN to detect similar binding sites [32]. Another 3D-based predictor is DeepSite, which deploys a CNN to binding pockets represented as voxels annotated with various atomic-based pharmacophoric properties [33].

In addition to the Euclidean space, many contemporary data, such as social networks, sensor networks, biological networks, and meshed surfaces, have an underlying structure that belongs to the non-Euclidean domain. Graph neural network (GNN) is a group of deep learning models designed to work specifically with non-Euclidean graph data [34]. GNNs have been demonstrated to achieve unparalleled performance in numerous applications against non-Euclidean data, including text classification [35], traffic prediction [36], and complex physics simulations [37]. GNNs were deployed to address important problems in biology as well, for instance, to predict the quantum properties of organic molecules [38], generate molecular fingerprints [39], detect protein interfaces [40], and identify drug-target interactions [41]. These applications are based on a notion that molecular structures can conveniently be represented as graphs, in which atoms are nodes, and chemical bonds are undirected edges connecting pairs of nodes.

In this communication, we expand the repertoire of graph-based approaches in biology and biomedicine by developing GraphSite, a new method to classify ligand binding sites with a GNN. First, a large and diverse dataset of binding sites are converted into graphs preserving the physicochemical properties of local protein structures, which are then used to train a GNN classifier. In contrast to computationally more intensive methods operating in the Euclidean space, lightweight GraphSite generates the graph representations of ligand binding site on-the-fly without any pre-processing requirements. Encouragingly, it not only achieves state-of-the-art performance in multi-class classification benchmarks with respect to other approaches but also generalizes well to unseen data. A comprehensive analysis of selected predictions by GraphSite demonstrates that its high performance is a result of the ability to effectively learn the underlying patterns of various types of binding pockets. We would like to note that the current GraphSite employing a GNN model to classify ligand binding sites is distinct from another software with the same name that utilizes a graph transformer to predict DNA binding residues in protein structures [42].

## 2. Materials and Methods

### 2.1. Datasets of Ligand Binding Pockets

A non-redundant collection of 51,677 pockets were compiled in September 2019 following a protocol developed previously to construct a dataset to evaluate binding site prediction with *e*FindSite [7,43]. Binding ligands in the *e*FindSite dataset were clustered at a Tanimoto coefficient (TC) threshold of 0.7 with the SUBSET program [44]. The 30 most abundant clusters were then manually curated into 14 pocket classes, referred to as the benchmarking dataset. The benchmarking dataset was divided into training (80%) and testing (20%) subsets by randomly splitting each class at a 4:1 ratio. The unseen dataset was created by selecting ligand-bound protein structures deposited to the Protein Data Bank (PDB) [45] no earlier than October 2019. Those proteins having a sequence identity of ≥50% to any protein in the benchmarking dataset were excluded. Pocket classes were assigned based on the chemical similarity of binding ligands to small molecules in the benchmarking dataset at a TC threshold of 0.7. This procedure resulted in 45 unseen pockets assigned to 9 classes. Finally, as the negative dataset, we use a previously published collection of 42 surface pockets resembling binding sites but not known to bind any ligand [46].

The *e*FindSite collection of ligand binding pockets [7,43] was first clustered by ligand chemical similarity and then the 30 most abundant clusters were manually curated into a dataset of 14 pocket classes. Clusters containing ATP, adenosine diphosphate (ADP), phosphoaminophosphonic acid-adenylate ester (ANP), uridine monophosphate (UMP), thymidine monophosphate (TMP), nicotinamide adenine dinucleotide, adenosine, azamethionine-5′-deoxyadenosine, and β-D-erythrofuranosyl adenosine, were merged to form class 0 (nucleotides). Further, clusters composed of glucose, fructose, α-D galactopyranose, and manopyranose, were combined into class 2 (carbohydrates). Another merged class 5 comprises phosphocholine, bromododecanol, tetradecylpropanedioic acid, oleic acid, palmitic acid, and hexaenoic acid. Clusters containing amino acids, such as lysine, arginine, and norvaline, citric acid and its derivatives, tartaric acid, tetraglycine phosphinate, and 1,3 dihydroxyacetone phosphate were joined to class 6. Finally, class 10 includes methylbenzamide, pentanamide, hexaethylene glycol, and tetraethylene glycol. The remaining clusters were sufficiently distinct to become separate classes. The clustering procedure followed by a manual data curation resulted in the benchmarking dataset of 21,124 pockets assigned to 14 classes binding a variety of ligands listed in Table 1.

### 2.2. Graph Representation of Binding Sites

Ligand binding pockets are converted to graphs, which are the input for the classifier. The nodes of these graphs are atoms contacting ligands identified through the analysis of interatomic contacts with the Ligand-Protein Contacts (LPC) software [47]. Nodes are connected by undirected edges when the distance between two atoms is ≤4.5 Å. We employ 11 node features, 7 of which are spatial features, and the other 4 are physicochemical/evolutionary features. Spatial features defining the shape of binding pockets include atomic Cartesian coordinates (x,y,z), spherical coordinates (r,ϑ,γ), and the solvent accessible surface area (SASA). Physicochemical/evolutionary features comprising charge, hydrophobicity, binding probability, and sequence entropy have been previously used in Bionoi, a method to represent ligand binding sites as Voronoi diagrams [48]. To distinguish between various bonding and non-bonding interactions, the bond multiplicity is used as the edge attribute with the value of 1.5 for aromatic bonds and 0 for non-covalent interactions.

Figure 1 illustrates the procedure to transform pockets into graphs. Atoms of binding residues become nodes connected to neighboring nodes within a distance threshold of 4.5 Å. To distinguish between bonding and non-bonding interactions, the edge attribute is set to either the bond multiplicity if two atoms form a chemical bond or 0 for those atoms interacting non-covalently. Individual nodes are assigned two types of features, spatial features defining the shape of the binding pocket (atomic coordinates and the solvent accessible surface area) and physicochemical/evolutionary features describing various properties, such as the charge, the hydrophobicity, the binding probability, and the sequence entropy. Representing pockets as graphs captures their overall characteristics and enables the information flow between atoms during the GNN model training.

### 2.3. Graph Neural Network

As pockets are represented as graphs, the binding site classification task becomes a graph classification problem essentially. A general graph classification framework employing a GNN incorporates three key components, message passing, the graph readout, and the classification stage. The overall architecture of a classifier implemented in GraphSite is presented in Figure 2. The main module consists of an embedding network (Figure 2B–D) comprising message passing layers (Figure 2B), the jumping knowledge connections (Figure 2C), and a global pooling layer to perform the graph readout (Figure 2D). As illustrated in Figure 2B, the node features of the input graph are first iteratively updated by neural weighted message (NWM) passing layers $h_{\omega }$ taking the edge attribute $e_{12}$ as input to generate $a_{12}$ as the weight of a message propagating from node 2 to node 1. Subsequently, the jumping knowledge network (JK-Net) [49] connecting message passing layers is employed, allowing the model to learn the optimal number of layers for individual nodes. The generated outputs are then processed by a max pooling layer performing a feature-wise pooling. The max pooling layer is followed by a global pooling layer to reduce the node feature dimension to a fixed-size vector, which is passed to a set of fully connected layers to generate the final classification result (Figure 2E).

#### 2.3.1. Message Passing

The role of message passing layers of the GNN is to update node features by propagating the information along edges. Node features updated with the information aggregated from neighbors contain valuable local patterns. Message passing layers in GraphSite adopt the general form of the neighborhood aggregation [50]:(1)$$x_{i}^{\left(k\right)}=\lambda \left(x_{i}^{\left(k-1\right)},\;\underset{j\in N\left(i\right)}{aggr}\phi \left(x_{i}^{\left(k-1\right)},\;x_{j}^{\left(k-1\right)},\;e_{ij}\right)\right),$$
where ϕ is a differentiable function generating a message, aggr is a permutation-invariant function aggregating all messages, and λ is the updating function. Other parameters are $x_{i}^{\left(k\right)}$ corresponding to the output feature vector of node i in layer k, $x_{j}^{\left(k\right)}$ representing feature vectors of the neighbors of node i, and the edge attribute $e_{ij}$. To better exploit node and edge features of binding site graphs, we implemented the following single-channel NWM:(2)$$x_{i}^{\left(k\right)}=h_{\theta }\left(\left(1+\epsilon \right)\cdot x_{i}^{\left(k-1\right)}+\sum_{j\in N\left(i\right)}h_{\omega }\left(e_{ij}\right)\cdot x_{j}^{\left(k-1\right)}\right),$$
where $h_{\omega }$ is an MLP taking the edge attribute as the input and outputting a message weight, which is a node feature *j*, ϵ is a learnable scalar, and $h_{\theta }$ is another MLP updating the aggregated information. Edge attributes are the same for all layers and are not updated during training. The NWM message passing rule can be regarded as an extension of the graph isomorphism network (GIN) [51], an expressive message passing model that is as powerful as the Weisfeiler–Lehman test in distinguishing graph structures. Its *sum* aggregator is replaced in GraphSite by the sum of weighted messages with weights generated by a neural network $h_{\omega }$. From another perspective, the NWM model belongs to the message passing neural network (MPNN) family [38]. The gated graph neural network (GGNN) is an MPNN family member whose message is formed by $A_{e_{ij}}x_{j}^{\left(k\right)}$, where $A_{e_{ij}}$ is a square transformation matrix generated by a multilayer perceptron (MLP) from the edge attribute $e_{ij}$. The GGNN can be regularized to the NWM by imposing a restriction on the matrix $A_{e_{ij}}$ to make it diagonal with all elements on the diagonal equal. We found empirically that the regularization of GGNN to NWM is not only computationally more efficient but also helps mitigate model overfitting.

Finally, inspired by the idea that multiple aggregators can improve the expressiveness of GNNs [52], we extended the single-channel NWM layer described by Equation (2) to a multi-channel NWM layer by concatenating the outputs of multiple aggregators:(3)$$x_{i}^{\left(k\right)}=h_{\theta }\left(\underset{c\in Channels}{concat}\left(\left(1+\epsilon_{c}\right)\cdot x_{i}^{\left(k-1\right)}+\sum_{j\in N\left(i\right)}h_{\omega c}\left(e_{ij}\right)\cdot x_{j}^{\left(k-1\right)}\right)\right),\;$$
where $\epsilon_{c}$ and $h_{\omega c}$ represent an aggregator learned as channel *c*. The aggregated node features are concatenated in their last dimension so that the concatenated node features have the shape of n by d×|C|, where d is the dimension of node features. The updated neural network $h_{\theta }$ also acts as a reduction function, decreasing the size of node features from d×|C| to d. Intuitively, the concatenation of multiple aggregators in the GNN is analogous to using multiple filters in the CNN; each aggregator corresponds to a filter, and the concatenated output is equivalent to the output feature maps in the convolution layer of the CNN.

#### 2.3.2. Graph Readout

A graph readout function reduces the size of a graph to a single node. GraphSite employs Set2Set [53] as a global pooling function to perform graph readout. Set2Set generates fixed-sized embeddings for sets of various sizes by utilizing the attention mechanism to compute the global representation of a set. Briefly, a long short-term memory (LSTM) [54] neural network recurrently updates a global hidden state of the input set. During the recurrent process, the global hidden state is used to compute attention values associated with each element in the set, which are in turn used to update the global hidden state. After several iterations, a global graph representation is created by concatenating the global hidden state constructed by the LSTM and the weighted sum of elements in the set. The global pooling layer reduces the node feature dimension from n×d to d, where n is the number of nodes and d is dimension of the node feature vector.

#### 2.3.3. Loss Function

The dataset of ligand binding pockets is imbalanced, meaning that some classes, such as nucleotide, have many more data points than other classes. Consequently, a training mini batch contains mostly the data from major classes, which could bias a typical loss function utilizing the cross-entropy. To mitigate this problem, GraphSite employs the focal loss (FL) function adding a damping factor ${\left(1-p_{t}\right)}^{\gamma }$ to the cross-entropy loss [55]:(4)$$FL\left(p_{t}\right)=-{\left(1-p_{t}\right)}^{\gamma }\log\left(p_{t}\right),$$
where $p_{t}$ is the predicted probability generated by the softmax function, and γ≥0 is a tunable hyperparameter. With this damping factor, dominating predictions with high probabilities are suppressed, while those predictions having low probabilities are assigned higher weights. This approach has been shown to minimize the problem of imbalanced classes.

### 2.4. Other Methods to Classify Pockets

A docking-based approach employs a small library of 14 ligands, each representing one class of pockets listed in Table 1. These compounds are docked to a query pocket with a molecular docking program smina [56] and the class of a molecule with the best docking score is assigned to that pocket. A pocket matching-based approach scans a query pocket against a small library of 14 representative pockets for all classes in Table 1 with a local structure alignment program G-LoSA [57]. The query pocket is then assigned a class from the library pocket having the best matching score. A random classifier randomly assigns the query pocket with a class according to the frequencies of individual classes in the dataset.

## 3. Results

### 3.1. Classification Performance against the Benchmarking Dataset

The performance of GraphSite is compared to that of several other approaches, GIN, molecular docking, pocket matching, and a random classifier. The GIN is an expressive message passing model, shown to be as powerful as the Weisfeiler–Lehman algorithm in distinguishing graph structures [51]. As the GIN employs a sum aggregator ignoring edge attributes, it constitutes an appropriate baseline to demonstrate the benefit of taking advantage of edge attributes in GraphSite with the NWM model. To conduct a fair comparison, the configurations of GraphSite and GIN are identical, except for the architecture of GNN layers. In addition to GNN-based classifiers, we also include docking- and pocket matching-based approaches. The former method employs smina [56], a fork of AutoDock Vina [58] featuring improved scoring and minimization, whereas pocket matching is conducted with G-LoSA, a tool to align protein local structures in a sequence order independent way [57].

After training, Graphsite and GIN is tested on the testing split of the dataset. Training the GraphSite classifier on Nvidia V100 GPU for 200 epochs took about 5 h. The classification performance of all tested methods on the testing subset is reported in Table 2. GraphSite achieves the best overall classification accuracy with a high recall of 81.3% and F1-score of 81.7%. Both recall and F1-score for the GIN are lower, therefore, utilizing edge attributes with multi-channel NWM layers indeed improves the classification accuracy over GIN layers. The performance of docking- and pocket matching-based approaches assessed by the recall and F1-score is comparable to that of a random classifier. Despite this low sensitivity, both techniques achieve relatively high precision, corresponding to a high fraction of correctly classified instances among all pockets. We note that docking and pocket matching were executed with default parameters because it is impractical to apply these algorithms exhaustively to increase the classification accuracy further.

Figure 3 shows the confusion matrix calculated for GraphSite predictions against the benchmarking dataset, in which numbers on the diagonal are recall values for ligand classes. Although GraphSite correctly predicted most classes, it misclassified a few pockets as well. There are two main reasons for these misclassifications. First, the support for some pocket classes across the dataset is low; for instance, only 1.8% of instances belong to class 12 and 1.6% to class 13 (Table 1). As more gradients are generated for the majority of classes during training, the model learns these classes more efficiently. Although this issue can partially be mitigated by employing the focal loss [55], the performance of minority classes is still going to be somewhat lower compared to those classes having stronger support. The second reason is that ligands binding to pockets belonging to different classes can, in fact, contain similar chemical moieties. We discuss several representative examples of these misclassifications in the following section.

### 3.2. Examples of Misclassified Pockets

Class 12 comprises pockets binding ligands containing morpholine rings, 17% of which are misclassified as nucleotides (Figure 3). Examples of these molecules are commonly used organic buffering agents [59], such as piperazine-N,N′-bis(2-ethanesulfonic acid) (PIPES). GraphSite classified a binding site in centromere-associated protein E (CENP-E) complexed with PIPES (PDB-ID: 1t5c) [60] as a nucleotide-binding pocket with a confidence score of 0.96. This prediction can be validated by structurally aligning the CENP-E pocket with a known nucleotide binding site. Here, we selected the ATP binding site in phosphoribosylformylglycinamidine (FGAM) synthase II (PDB-ID: 2hs0) [61], whose sequence identity with CENP-E is only 21%. Ligand binding sites in both proteins were aligned with PocketAlign, which employs shape descriptors in the form of geometric perspectives, supplemented by chemical group classification, to compute sequence order-independent alignments [62]. Figure 4A shows the superposition of binding sites in CENP-E (purple) and FGAM synthase II (yellow). Encouragingly, the root-mean-square deviation over Cα atoms (Cα-RMSD) of 9 equivalent residues is as low as 1.6 Å. Generally, values below 3.0 Å indicate that the aligned pockets are structurally similar [62].

Another example is 2-(N-morpholino)ethanesulfonic acid (MES) containing the morpholine ring that is structurally related to the piperazine ring with one nitrogen atom replaced by oxygen [63]. GraphSite classified a binding pocket in zinc transport transcriptional regulator (zitR) complexed with MES (PDB-ID: 5yhz) [64] as a nucleotide-binding pocket with a confidence score of 0.97. Figure 4B shows that this pocket (purple) is structurally related to the ATP binding site in FGAM synthase II (yellow) with 1.5 Å Cα-RMSD over 6 equivalent residues reported by PocketAlign. Note that the global sequence identity between zitR and FGAM is only 20%. Piperazine and morpholine rings are often used to develop molecules competing with nucleotides. For instance, morpholinos, nucleotide analogs blocking mRNA splicing and translation [65], contain the morpholine ring replacing the sugar group of a nucleotide [66]. Further, morpholine-containing pyrazolopyrimidines are selective and potent ATP-competitive inhibitors of mTOR, showing anti-cancer properties in xenograft tumor models [67]. ATP-competitive inhibitors often contain piperazine rings to increase their aqueous solubility [68] and to form favorable interactions with the hinge region of protein kinases [69].

An example of the ATP-competitive inhibitor containing piperazine is imatinib, a widely used chemotherapeutic to treat certain types of cancer [70]. Piperazine and benzene rings in imatinib are required for their inhibitory activity against leukemia cell lines [10]. A binding site in Src-Abl tyrosine kinase ancestor (ANC-AS) complexed with imatinib (PDB-ID: 4csv) [71] was classified by GraphSite as a nucleotide-binding pocket with a confidence of 0.99. Despite a low sequence identity between ANC-AS and FGAM synthase II of 23%, PocketAlign aligned their binding sites with a Cα-RMSD of 1.8 Å over 17 equivalent residues (Figure 4C, ANC-AS is purple and FGAM yellow), indicating that both pockets can bind similar ligands. Indeed, ANC-AS has also been co-crystallized with ATP (PDB-ID: 4ueu); therefore, the classification by GraphSite is, in fact, correct. This is an example of a pocket capable of binding multiple, chemically dissimilar ligands, which may belong to more than one class.

GraphSite classified 26% of pockets binding alkyl phosphates belonging to class 13 as binding sites for essential amino acids (Figure 3). For instance, a binding site in a coenzyme A-dependent thiolase LsrF bound to (3R)-3-hydroxy-2,4-dioxopentyl dihydrogen phosphate (PDB-ID: 4p2v) [72] was classified as an essential amino acid binding pocket with 0.96 confidence. Figure 4D shows a valid structure alignment constructed by PocketAlign between this pocket (purple) and a known amino acid binding pocket in L-arginine:glycine amidinotransferase (AT, yellow) complexed with arginine (PDB ID:4jdw) [73]. This alignment has a Cα-RMSD of 1.5 Å calculated over 14 equivalent residues indicating that the binding site in LsrF is structurally related to arginine binding pockets. As a matter of fact, alkyl phosphates and amino acids are connected through common biochemical pathways, e.g., phosphoenol pyruvate is an important citric acid cycle intermediate that produces alpha-ketoglutarate, ultimately leading to the synthesis of amino acid arginine [74,75]. This may explain the classification result by GraphSite of the binding site in LsrF.

Colchicine is an anti-inflammatory agent primarily used to treat gout [76]. A colchicine binding site in human bromodomain-containing protein 4 (BRD4, PDB-ID: 6ajz) [77] was classified by GraphSite as a nucleotide binding site with a confidence score of 0.93. Interestingly, BRD4 is homologous to the murine mitotic chromosome-associated protein [78] and the human RING3 protein [79], both annotated with kinase activity. Colchicine is also effective against acute coronary syndrome by inhibiting a nucleotide-binding domain (NOD)-like receptor protein 3 inflammasome protein complex [80]. The colchicine binding site in BRD4 was aligned to a known ATP binding site in FGAM synthase II with PocketAlign. The resulting alignment shown in Figure 4E has a low Cα-RMSD of 1.7 Å over 9 equivalent residues (BRD4 is purple and FGAM is yellow). This result indicates that both pockets are structurally similar, explaining the classification by GraphSite of the pocket in BRD4 as nucleotide binding.

A few pockets binding essential amino/citric/tartaric acids belonging to class 6 were classified by GraphSite as binding sites for lipids (Figure 3). An example is a pocket in maltose O-acetyltransferase from *E. coli* binding tromethamine (MAT, PDB-ID: 6ag8) [81] assigned by GraphSite to class 5 with a confidence score of 0.98. MAT catalyzes the CoA-dependent transfer of an acetyl group to maltose and other sugars [82]. The fatty acid or lipid biosynthesis pathway produces acetyl CoA that enters the citric acid cycle to produce citrate [83]. According to results by PocketAlign shown in Figure 4F, the binding site in MAT is structurally similar to a pocket in putative endonuclease/exonuclease/phosphatase family protein binding di(hydroxyethyl)ether (BtR318A, PDB-ID: 3mpr) [84] with an RMSD of 1.5 Å over 8 equivalent residues (MAT is purple and BtR318A is yellow). This high similarity to a lipid-binding site gives a reason for the misclassification of a pocket in MAT by GraphSite.

### 3.3. Performance on Unseen Data

Next, the performance of GraphSite is evaluated against a small dataset of “unseen” pockets. All data in this set were published later than the benchmarking dataset; thus, these pockets have not been used to train the machine learning model. In addition, the unseen dataset comprises only those proteins having low homology to benchmarking proteins. Encouragingly, using GraphSite yields the weighted recall, precision, and F1-score against the unseen dataset of 68.9%, 75.5%, and 70.7%, respectively. Although these values are somewhat lower than those reported in Table 2, the performance of GraphSite is still satisfactory considering that the unseen dataset is smaller and much more challenging than the benchmarking dataset. GraphSite is expected to achieve such performance in real-world applications employing new data.

### 3.4. Classification of the Negative Dataset

Lastly, GraphSite was applied to the negative dataset of surface pockets having characteristics of binding sites yet not binding any ligands [46]. Figure 5 shows that the distribution of the classification confidence is diametrically different from that obtained for the benchmarking dataset. A purple violin plot on the left shows the distribution of the probability of the top-ranked class predicted by GraphSite for the benchmarking dataset. The median probability of 0.93 indicates that the model produced not only accurate but also highly confident predictions for the benchmarking dataset. Note that this performance was obtained employing a proper cross-validation protocol. In contrast, predictions for the negative dataset are clearly less confident, with a median probability of only 0.67. These results demonstrate that even though non-binding sites were classified into 14 classes as GraphSite was designed for, unconfident predictions indicate that these surface pockets do not fit well any ligand class the model was trained against.

### 3.5. Siamese-GraphSite Extension

In addition to the classifier model, we extended GraphSite by adding a Siamese model for metric learning. This model generates two graph embeddings for a pair of input graphs, which are then used to calculate the contrastive loss (CL) [85]:(5)$$CL\left(W,y,x_{1},x_{2}\right)=\frac{1}{2}\left(1-y\right){\left(d_{W}\right)}^{2}+\frac{1}{2}\left(y\right){\left(max\left(0,m-d_{W}\right)\right)}^{2}$$
where y is the label of the pair of input graphs $x_{1}$ and $x_{2}$ (either 1—similar or 0—dissimilar), W parameterizes the embedding network, $d_{W}$ is the Euclidean distance between graph embeddings, and m>0 is a distance margin for the input pair to contribute to the loss function. Intuitively, using the contrastive loss in model training results in embeddings from the same class being close to one another in the Euclidean space and far away from each other for embeddings belonging to different classes.

As shown in Figure 6, embedding networks with shared parameters require a pair of graphs representing binding pockets as the input to generate two graph embeddings. These embeddings can subsequently be used in various machine learning applications, such as the visualization of the binding pocket conformational space. As this architecture optimizes the relative distances of the data in the Euclidean space, embeddings generated by Siamese-GraphSite are well suited for distance-based analyses, including, for instance, t-distributed stochastic neighbor embedding (t-SNE) visualization [86] and *k*-nearest neighbor clustering [87].

To test the distance metric learning on weakly supervised data, we trained Siamese-GraphSite against 8 clusters in the original dataset prior to the manual curation. Figure 7 shows the t-SNE visualization of the clusters from the validation subset (10%) after the model was trained on the remaining subset (90%). Overall, similar pockets are grouped together, while dissimilar pockets are located away from one another. Interestingly, clusters 0 (green dots in Figure 7) and 3 (orange dots in Figure 7) come together according to the t-SNE analysis. The former cluster contains ADP and ANP, whereas the latter is composed of UMP and TMP. Because of the functional similarity of pockets belonging to these clusters, both groups were merged during the manual curation of the dataset into a single class 0 comprising nucleotides (Table 1). Similarly, clusters 3 (red dots in Figure 7) and 8 (yellow dots in Figure 7) are grouped together. These clusters containing glucose and fructose ligands were also manually curated into a single class 2 composed of carbohydrates (Table 1). These observations indicate that the Siamese model effectively learns embeddings to represent functional relations among binding pockets in line with the human expert knowledge.

## 4. Discussion

In this communication, we describe GraphSite, a method to classify ligand binding sites, represented as graphs, with a graph deep learning model. Comprehensive benchmarking calculations demonstrate that the trained classifier extracts informative features of binding pockets yielding state-of-the-art classification performance. Importantly, GraphSite successfully classifies binding sites without any information on their ligands. It has the desired capability to generalize to unseen data, as shown for an independent dataset of pockets taken from proteins having low homology and solved posterior to training structures. Moreover, calculations conducted for the negative dataset of surface pockets not binding any ligands demonstrate that GraphSite does not overpredict; therefore, the false positive rate in real applications should be low.

GraphSite can further be extended in several directions. Utilizing larger datasets comprising more classes will not only help train a more powerful and accurate classifier, but it will also increase the performance of metric learning by the Siamese model presented here as an example of the extension of GraphSite. However, this plan of action would require employing various data augmentation techniques [31] to account for fewer structures currently available for certain pocket classes. We also expect that exploring additional node features of binding site graphs may also improve the classification performance. GraphSite is a versatile approach that can be useful in other deep learning-based applications involving the analysis of ligand binding sites. For example, it is possible to train a graph autoencoder to generate latent embeddings of binding sites for subsequent use in machine learning. Another potential application is to build a model to predict drug-target interactions where the GNN layers of GraphSite can be used as the feature extractor for input binding sites. These new directions to improve and extend GraphSite will be explored in the future.

## Figures and Tables

**Figure 1 biomolecules-12-01053-f001:**
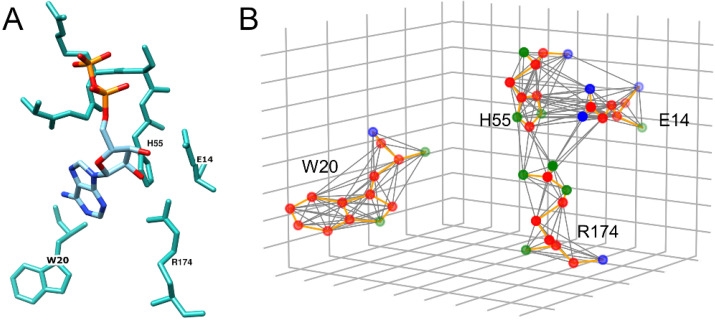
Example of the graph representation of a binding site. (**A**) The structure of a binding pocket for ADP in DnaA regulatory inactivator Had from *E*. *coli* (PDB-ID: 5x06). (**B**) The graph representation of four residues, W20, R174, E14, and R53, selected from (**A**).

**Figure 2 biomolecules-12-01053-f002:**
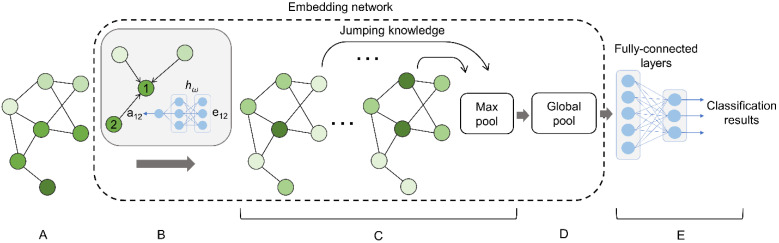
Architecture of the pocket classifier in GraphSite. (**A**) The input graph represents a binding site. (**B**) A neural network computing the weight for message passing from the edge attributes of the input graph. (**C**) Message passing layers of the jumping knowledge network. (**D**) A global pooling layer implementing the Set2Set model. (**E**) Fully connected layers generate the final classification results.

**Figure 3 biomolecules-12-01053-f003:**
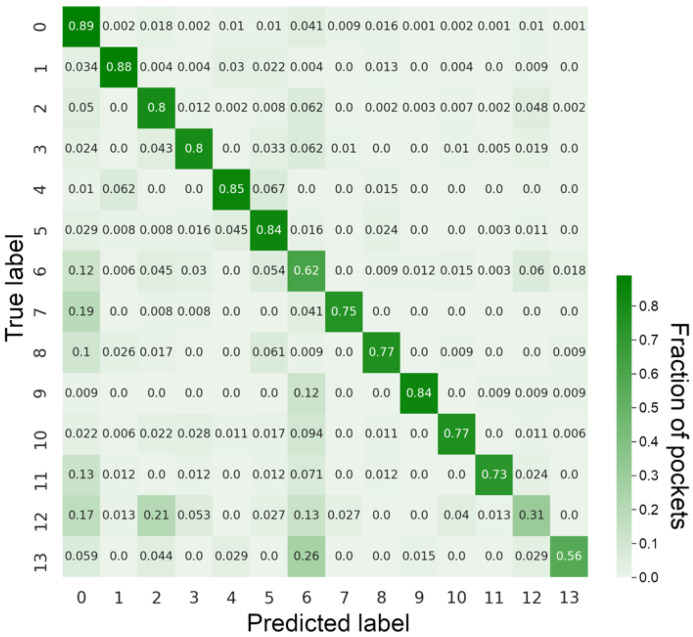
Confusion matrix for classification with GraphSite on the benchmarking dataset. Each row of the confusion matrix is normalized. Numbers on the diagonal correspond to the recall of each class, while other numbers indicate the fraction of misclassified pockets.

**Figure 4 biomolecules-12-01053-f004:**
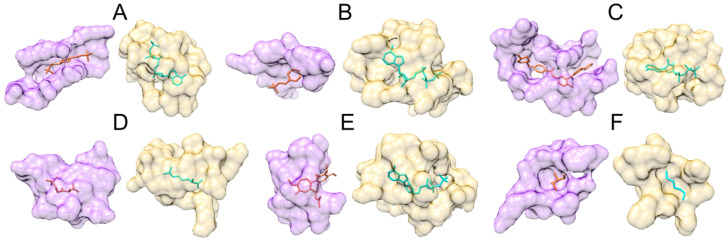
Structure alignments between misclassified pockets and those belonging to the predicted class. (**A**) PIPES (orange sticks) binding site in CENP-E (purple surface) and ATP (cyan sticks) binding site in FGAM synthase II (yellow surface). (**B**) MES (orange sticks) binding site in zitR (purple surface) and ATP (cyan sticks) binding site in FGAM synthase II (yellow surface). (**C**) Imatinib (orange sticks) binding site in ANC-AS (purple surface) and ATP (cyan sticks) binding site in FGAM synthase II (yellow surface). (**D**) (3R)-3-hydroxy-2,4-dioxopentyl dihydrogen phosphate (orange sticks) binding site in LsrF (purple surface) and arginine (cyan sticks) binding site in AT (yellow surface). (**E**) Colchicine (orange sticks) binding site in BRD4 (purple surface) and ATP (cyan sticks) binding site in FGAM synthase II (yellow surface). (**F**) Tromethamine (orange sticks) binding site in MAT (purple surface) and di(hydroxyethyl)ether (cyan sticks) binding site in BtR318A (yellow surface).

**Figure 5 biomolecules-12-01053-f005:**
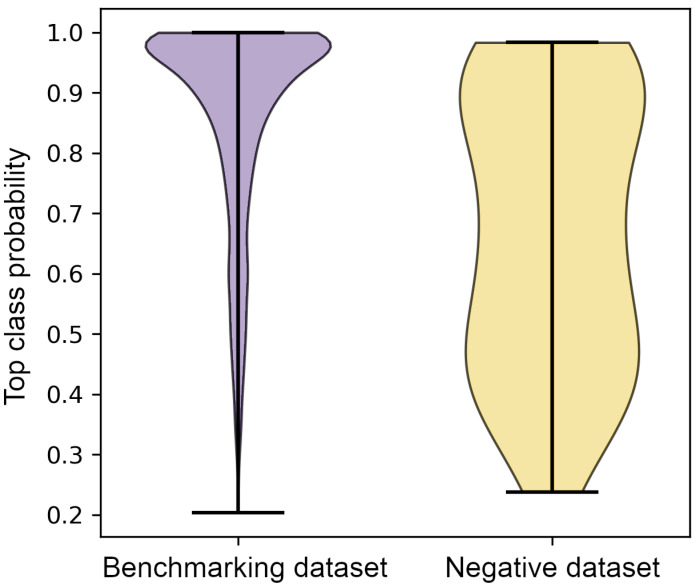
Distribution of the classification confidence for benchmarking and negative datasets. The classification confidence corresponds to a probability of the top-ranked ligand binding class predicted by GraphSite.

**Figure 6 biomolecules-12-01053-f006:**
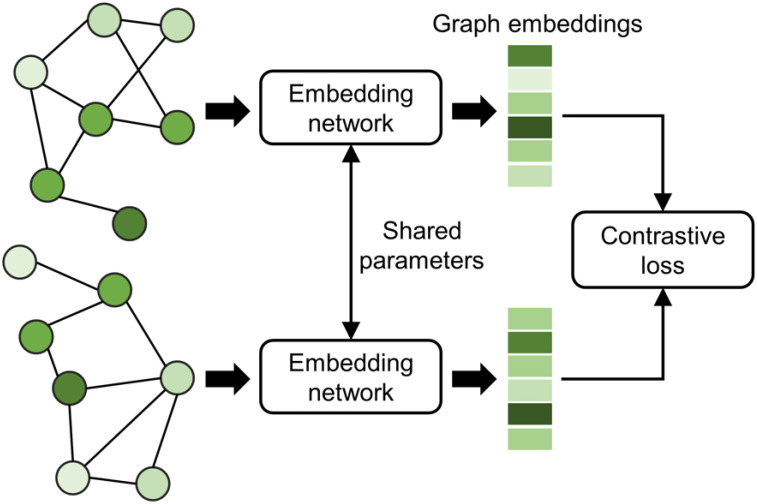
Architecture of Siamese-GraphSite. This model requires a pair of graph-structured data as the input for two embedding networks sharing their parameters and utilizes the contrastive loss function.

**Figure 7 biomolecules-12-01053-f007:**
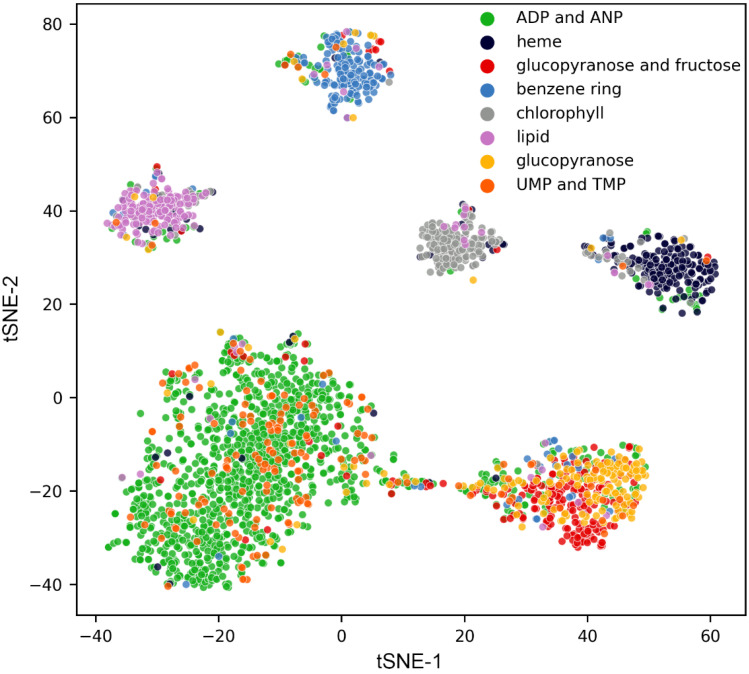
t-SNE visualization of embeddings generated by Siamese-GraphSite. Each dot represents one pocket colored by the cluster assignment.

**Table 1 biomolecules-12-01053-t001:** Classes of ligand binding sites in the primary benchmarking dataset. Support is the number of pockets in the dataset.

Class	Binding Ligands	Support
0	nucleotide	7625
1	heme	1158
2	carbohydrate	3001
3	benzene ring	1054
4	chlorophyll	968
5	lipid	1890
6	essential amino/citric/tartaric acids	1663
7	S-adenosyl-L-homocysteine	602
8	coenzyme A	573
9	pyridoxal phosphate	566
10	benzoic acid	897
11	flavin mononucleotide	417
12	morpholine ring	374
13	phosphate	337

**Table 2 biomolecules-12-01053-t002:** Classification performance against the benchmarking dataset. GraphSite is compared to the graph isomorphism network (GIN), molecular docking with smina, pocket matching with G-LoSA, and a random classifier. Precision, recall, and F1-score are class-weighted.

Method	Recall	Precision	F1-Score
GraphSite	81.3%	82. 3%	81.7%
GIN	75.1%	74.3%	74.3%
Smina	16.7%	43.4%	16.1%
G-LoSA	14.8%	34.4%	15.9%
Random	17.8%	17.7%	17.7%

## Data Availability

GraphSite is available at https://github.com/shiwentao00/Graphsite-classifier and datasets are available at https://osf.io/svwkb/, accessed on 18 July 2022.

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
