# Peer review of "GraphSite: Ligand Binding Site Classification with Deep Graph Learning"

_biomolecules, 2022, doi:10.3390/biom12081053_

Round 1

Reviewer 1 Report

The authors describe a modern approach to analyze protein structures and find ligand binding pockets in them. Accompanying software is available and seems to be well documented (although I have to admit that I did not try it). The idea to use graphs to describe interactions in molecules (proteins and smaller organic ligands) is reasonable, as graphs naturally describe the molecules and molecular interactions.

Although I have some questions regarding the wide applicability of the described GraphSite method, the ideas are worth publishing. However, in my opinion, some questions need to be clarified first.

General comments to the presentation and questions:

* The description of research starts with the "Results" section directly after Introduction. As a result, it would be nice to see some experimental details repeated at least shortly in the results description. This would help the readers to understand the conducted research without hopping forward and back to the "Methods" section in the end. For example, in the beginning it is not clear what features of the atoms are used in the binding site graph (line 122).

* I did not understand whether the binding site graph has only atoms or residues contacting directly with the ligand, or it does go further into the protein structure? In principle, the whole protein structure graph could be used as a binding site graph having the appropriate labeling of atoms and residues, so I did not get how far from the ligand does the binding site description stop?

* I liked the discussion about mispredictions very much, because in reality some drug-like ligands are designed to bind to the same binding sites as biological coenzymes or cofactors.

* In similar modeling works getting the negative examples for classification is usually quite hard, because only positive cases of ligand binding sites are observed experimentally in the protein structures. The authors avoid this problem by using only positive examples belonging to 14 classes, and by developing a multi-class classifier. First, I would suggest to describe this more clearly in the beginning of the result section, as usually "classification problem" can be understood as two-class classification problem. Yet more interesting is the question, what would the model predict for the really negative cases, i.e., for the proteins that do not bind any of the ligands from the 14 investigated classes. I guess that there would be a lot of false positive predictions. Maybe the authors could discuss this questions in more detailed or even conduct a short prediction experiment?

* The splitting into training and test sets before machine learning is not described, therefore I assume that the results in table 2 probably are for the training set and are largely overfitted. As a result, it is not very fair to compare to other methods that have not been trained using the same training set. I think that this should be clarified at least in the text, if not taken into account during the study design.

* Lines 207-210 state that other methods have been used with default parameters. I wonder, maybe more advanced usage of these methods (docking and protein structure alignment) could lead to better binding site prediction results?

Text-related comments:

* Line 45: I would suggest writing "despite the encouraging progress"

* In some places "binding sites" are called "biding sites" (N is missing)

* Line 328 "All data in this dataset have timestamps that are earlier than the latest timestamp in the benchmarking dataset" - is it really so? Probably in the unseen dataset the structures are published later than in the benchmarking dataset, not earlier?

Author Response

* The description of research starts with the "Results" section directly after Introduction. As a result, it would be nice to see some experimental details repeated at least shortly in the results description. This would help the readers to understand the conducted research without hopping forward and back to the "Methods" section in the end. For example, in the beginning it is not clear what features of the atoms are used in the binding site graph (line 122).

We follow “Instructions for Authors” for this journal, which state “We do not have strict formatting requirements, but all manuscripts must contain the required sections: Author Information, Abstract, Keywords, Introduction, Materials & Methods, Results, Conclusions, Figures and Tables with Captions, Funding Information, Author Contributions, Conflict of Interest and other Ethics Statements.” We certainly agree with the Reviewer that some methodological information should be provided in the “Results” section if it is placed before the “Methods” section. In response to this comment, we added the requested revisions:

Page 6 (top): 2.1. Graph representation of biding sites
“Individual nodes are assigned two types of features, spatial features defining the shape of the binding pocket (atomic coordinates and the solvent accessible surface area), and physicochemical/evolutionary features describing various properties, such as the charge, the hydrophobicity, the binding probability, and the sequence entropy.”

* I did not understand whether the binding site graph has only atoms or residues contacting directly with the ligand, or it does go further into the protein structure? In principle, the whole protein structure graph could be used as a binding site graph having the appropriate labeling of atoms and residues, so I did not get how far from the ligand does the binding site description stop?

In our work, binding site graphs are constructed from atoms contacting ligands. However, we would like to note that GraphSite is quite versatile and can accommodate any binding site definition. To clarify this issue, we added the following revisions:

Page 14 (bottom): 4.2. Graph representation of biding sites
“Nodes of these graph are atoms contacting ligands identified through the analysis of interatomic contacts with the Ligand-Protein Contacts (LPC) software [81].”

References
“81. Sobolev, V., et al., Automated analysis of interatomic contacts in proteins. Bioinformatics, 1999. 15(4): p. 327-32.”

* I liked the discussion about mispredictions very much, because in reality some drug-like ligands are designed to bind to the same binding sites as biological coenzymes or cofactors.

We absolutely agree with the Reviewer and are very grateful that he or she appreciates the discussion of mispredictions.

* In similar modeling works getting the negative examples for classification is usually quite hard, because only positive cases of ligand binding sites are observed experimentally in the protein structures. The authors avoid this problem by using only positive examples belonging to 14 classes, and by developing a multi-class classifier. First, I would suggest to describe this more clearly in the beginning of the result section, as usually "classification problem" can be understood as two-class classification problem. Yet more interesting is the question, what would the model predict for the really negative cases, i.e., for the proteins that do not bind any of the ligands from the 14 investigated classes. I guess that there would be a lot of false positive predictions. Maybe the authors could discuss this questions in more detailed or even conduct a short prediction experiment?

We very much appreciate this comment and agree that it would be interesting to analyze the classifier performance against negative cases as suggested by this Reviewer. In response to this comment, we ran GraphSite for a negative dataset of 42 surface pockets not binding any ligands. This dataset, published by the Sternberg group at Imperial College London [BMC Bioinformatics volume 13, Article number: 162 (2012)], is described in the original paper as “The non-binding sites are surface pockets that look like binding sites but are not known to bind any ligand.” The revised manuscript contains the following changes:

Page 12 (top): 2.7. Classification of the negative dataset
“Lastly, GraphSite was applied to the negative dataset of surface pockets having characteristics of binding sites, yet not binding any ligands [76]. Although these pockets were classified into 14 classes, Figure 5 shows that the distribution of the classification confidence is diametrically different from that obtained for the benchmarking dataset. A purple violin plot on the left shows the distribution of the probability of the top-ranked class predicted by GraphSite for the benchmarking dataset. The median probability of 0.93 indicates that the model made not only accurate, but also highly confident predictions for the benchmarking dataset. Note that this performance was obtained employing a proper cross-validation protocol. In contrast, predictions for the negative dataset are clearly less confident with the median probability of only 0.67. These results demonstrate that even though non-binding sites were classified into 14 classes as GraphSite was designed for, unconfident predictions indicate that these surface pockets do not fit well any class the model was trained against.”

Page 13 (middle): 3. Discussion
“Moreover, calculations conducted for the negative dataset of surface pockets not binding any ligands demonstrate that GraphSite does not overpredict, therefore the false positive rate in real applications should be low.”

Page 14 (middle): 4.1. Datasets of ligand binding pockets
“Finally, as the negative dataset, we use a previously published collection of 42 surface pockets resembling binding sites but not known to bind any ligand [76].”

References
“76. A Santos, J.C., et al., Automated identification of protein-ligand interaction features using Inductive Logic Programming: a hexose binding case study. BMC bioinformatics, 2012. 13(1): p. 1-11.”

* The splitting into training and test sets before machine learning is not described, therefore I assume that the results in table 2 probably are for the training set and are largely overfitted. As a result, it is not very fair to compare to other methods that have not been trained using the same training set. I think that this should be clarified at least in the text, if not taken into account during the study design.

Following the standards of best practices in the evaluation of machine learning, we report the performance on the testing set, not the training set. We agree with the Reviewer that this was unclear in the initial submission. The revised manuscript contains the following clarifications:

Page 8 (top): 2.4. Classification performance against the benchmarking dataset
“The classification performance of all tested methods on the testing subset is reported in Table 2.”

Page 14 (middle): 4.1. Datasets of ligand binding pockets
“The benchmarking dataset was divided into training (80%) and testing (20%) subsets by randomly splitting each class at a 4:1 ratio.”

* Lines 207-210 state that other methods have been used with default parameters. I wonder, maybe more advanced usage of these methods (docking and protein structure alignment) could lead to better binding site prediction results?

In our experience, executing these methods with default parameters yields their representative performance and changing parameters may result in a marginal improvement at most. Since this does not justify very high computational costs, we decided to employ default parameters.

Text-related comments:

* Line 45: I would suggest writing "despite the encouraging progress"

We have added the requested revision:

Page 3 (middle): 1. Introduction
“Despite the encouraging progress in ligand binding site detection, there is a need for a better functional characterization of the identified sites with respect to types and properties of binding molecules.”

* In some places "binding sites" are called "biding sites" (N is missing)

We are very grateful to this Reviewer for reading the text so carefully and pointing out these mistakes. We fix them in the revised version of the manuscript:

Page 5 (bottom): 2.1. Graph representation of binding sites
Page 14 (bottom): 4.2. Graph representation of biding sites

* Line 328 "All data in this dataset have timestamps that are earlier than the latest timestamp in the benchmarking dataset" - is it really so? Probably in the unseen dataset the structures are published later than in the benchmarking dataset, not earlier?

The Reviewer is right, and we revised the manuscript according to his or her suggestion:

Page 11 (bottom): 2.6. Performance on unseen data
“All data in this set were published later than the benchmarking dataset, thus these pockets have not been used to train the machine learning model.”

Reviewer 2 Report

Major:

Issue 1: The documentation of the tool is VERY poor. No information is given on how to generate the required input files (mol2 of the binding pocket, profile for sequence entropy and profile for solvent accessible surface area). The mol2 format is standard, but the two additional profiles are never explained, and no information is given on how to generate them. After digging around on my own through another of the group’s Github repositories (pocket2drug), I eventually found that the ASA profile must be generated using POPS (POPScomp), while the entropy profile must be generated using ProfilPro. No information about that is given in the github repository, or the manuscript. No example files are offered. This makes GraphSite impossible to use and evaluate. The authors should provide a much more detailed documentation and at least one set of example files. They should also provide instructions for an example run, including the steps required to generate the profiles needed.

Issue 2: The authors provide no information on the model’s performance or scaling capabilities. Considering the largely computational context of the work, as well as the computational resources typically required by deep learning methods, a benchmark of GraphSite’s performance and an overview of system requirements would be welcome.

Issue 3: Please provide a list for the PDB structures (PDB IDs and ligands evaluated) used in the validation of the tool. This can be a list in the paper (as supplementary material), or it can be added in the tool’s github repository.

Issue 4: The authors should consider implementing a web server version of the tool; this would help the dissemination of the method, especially for non-technical users.

Issue 5: a lot of the work presented here is actually part of another github repository (Graphsite-classifier), but that reprository is never referenced in the paper.

Minor:

Issue 1: Although the manuscript is written in good English, a few grammatical errors exist that need to be corrected.

Issue 2: A somewhat similar tool (for detecting DNA binding pockets based on AlphaFold structural predictions), also using the name GraphSite, already exists and has been published (doi: https://doi.org/10.1093/bib/bbab564 , github: https://github.com/biomed-AI/GraphSite ). The authors should make sure there are no trademark or copyright issues in using the name. 

Author Response

Issue 1: The documentation of the tool is VERY poor. No information is given on how to generate the required input files (mol2 of the binding pocket, profile for sequence entropy and profile for solvent accessible surface area). The mol2 format is standard, but the two additional profiles are never explained, and no information is given on how to generate them. After digging around on my own through another of the group’s Github repositories (pocket2drug), I eventually found that the ASA profile must be generated using POPS (POPScomp), while the entropy profile must be generated using ProfilPro. No information about that is given in the github repository, or the manuscript. No example files are offered. This makes GraphSite impossible to use and evaluate. The authors should provide a much more detailed documentation and at least one set of example files. They should also provide instructions for an example run, including the steps required to generate the profiles needed.

We agree with the Reviewer that the GraphSite software was not well documented. We updated the GitHub repository with more detailed instructions on how to prepare the required input files and run the codes. Instructions on using additional software are posted at https://github.com/shiwentao00/Graphsite-classifier/blob/master/docs/data_curation/readme.md. We also provide all input and output files from the Open Science Framework at https://osf.io/svwkb/. We are going to keep updating and improving GitHub pages for this project based on feedback from users.

Issue 2: The authors provide no information on the model’s performance or scaling capabilities. Considering the largely computational context of the work, as well as the computational resources typically required by deep learning methods, a benchmark of GraphSite’s performance and an overview of system requirements would be welcome.

GraphSite requirements are explained on the GitHub page. Training time depends on the specific hardware, however, since most users will be using the pretrained model, this should not be a concern. We appreciate this Reviewer’s comment and, therefore, we added the following revisions to provide the information on the training time on the state-of-the-art hardware:

Page 7 (top): 2.2. Graph neural network
“Training the GraphSite classifier on Nvidia V100 GPU for 200 epochs took about 5 hours.”

Issue 3: Please provide a list for the PDB structures (PDB IDs and ligands evaluated) used in the validation of the tool. This can be a list in the paper (as supplementary material), or it can be added in the tool’s github repository.

Following this comment, we made all data available from the Open Science Framework at https://osf.io/svwkb/. In response, we added the following revisions to the manuscript:

Abstract:
“GraphSite is available at https://github.com/shiwentao00/Graphsite and datasets are available at https://osf.io/svwkb/.”

Availability:
“GraphSite is available at https://github.com/shiwentao00/Graphsite and datasets are available at https://osf.io/svwkb/.”

Issue 4: The authors should consider implementing a web server version of the tool; this would help the dissemination of the method, especially for non-technical users.

We currently do not have a reliable and publicly available machine that can host the webserver. We are submitting grant proposals that include budget for such server. If any of these proposal gets funded, we will buy a new machine and set up a web server. Until then, other groups can download and run our codes locally.

Issue 5: a lot of the work presented here is actually part of another github repository (Graphsite-classifier), but that reprository is never referenced in the paper.

The GraphSite-classifier repository is linked from the main GraphSite repository referenced in the paper. We would prefer to keep the main GraphSite repository in the paper and link other resources from GitHub to avoid having too many URLs in the manuscript. We hope that this Reviewer will give us his or her permission to keep the current link structure.

Minor:

Issue 1: Although the manuscript is written in good English, a few grammatical errors exist that need to be corrected.

We proofread the manuscript and corrected several errors. Thank you for reading the manuscript so carefully and please let us know if we missed anything.

Issue 2: A somewhat similar tool (for detecting DNA binding pockets based on AlphaFold structural predictions), also using the name GraphSite, already exists and has been published (doi: https://doi.org/10.1093/bib/bbab564 , github: https://github.com/biomed-AI/GraphSite ). The authors should make sure there are no trademark or copyright issues in using the name.

Yes, this tool was published recently, after we created GraphSite repositories. It would be difficult to rename everything since our content has been available for some time now. Multiple tools having the same name can co-exist without any issues; the name does not seem to be copyrighted.

Round 2

Reviewer 1 Report

I would like to thank the authors who answered all my questions.

Author Response

Many thanks for your time and effort reviewing our paper.